# Niche partitioning facilitates coexistence of closely related honey bee gut bacteria

**Silvia Brochet[1], Andrew Quinn[1], Ruben AT Mars[2], Nicolas Neuschwander[1], Uwe Sauer[2], Philipp Engel[1]\***

[1]Department of Fundamental Microbiology, University of Lausanne, Lausanne, Switzerland; [2]Institute of Molecular Systems Biology, ETH Zürich, Zürich, Switzerland

**Abstract** Ecological processes underlying bacterial coexistence in the gut are not well understood. Here, we disentangled the effect of the host and the diet on the coexistence of four closely related *Lactobacillus* species colonizing the honey bee gut. We serially passaged the four species through gnotobiotic bees and in liquid cultures in the presence of either pollen (bee diet) or simple sugars. Although the four species engaged in negative interactions, they were able to stably coexist, both in vivo and in vitro. However, coexistence was only possible in the presence of pollen, and not in simple sugars, independent of the environment. Using metatranscriptomics and metabolomics, we found that the four species utilize different pollen-derived carbohydrate substrates indicating resource partitioning as the basis of coexistence. Our results show that despite longstanding host association, gut bacterial interactions can be recapitulated in vitro providing insights about bacterial coexistence when combined with in vivo experiments.

**\*For correspondence:**
philipp.engel@unil.ch

**Competing interests:** The authors declare that no competing interests exist.

## Introduction

Gut microbial communities are usually dominated by few bacterial phyla and families, but contain a high extent of species- and strain-level diversity (*Ley et al., 2008*; *Dethlefsen et al., 2007*). According to the competition-relatedness hypothesis, the more closely two organisms are related the more likely it is that they will compete and exclude each other due to overlapping niches (*Elton, 1946*). Therefore, it has remained unclear how closely related microbes can be maintained in the gut, or in any other natural microbial ecosystem.

The high concentration of nutrients and the structured environment of the gut may allow functionally redundant species or strains to coexist (*Ley et al., 2006*). The host may even select for such redundancy, as it can increase the stability and resilience of the microbiota against environmental disturbance (*Ley et al., 2006*; *Foster et al., 2017*). Phage predation can also contribute to the maintenance of diversity by imposing kill-the-winner dynamics and hindering the outgrowth of a single dominant strain (*Koskella and Brockhurst, 2014*). Another possibility is that closely related species, and even strains of the same species, have functionally diverged from each other and occupy distinct ecological niches (*Chesson, 2000*; *Bittleston et al., 2019*). The genomic flexibility of bacteria facilitates adaptation to different nutrients, provided in the diet or by the host (*Berasategui et al., 2017*; *Martens et al., 2008*), or result from interactions with other bacteria (*Madi et al., 2020*), such as cross-feeding (*Goldford et al., 2018*) or cooperative glycan breakdown (*Rakoff-Nahoum et al., 2016*).

Few experimental studies have investigated the coexistence of bacteria in host-associated microbial communities. The high diversity in these ecosystems and the resistance of many host-associated bacteria to experimental manipulations introduce considerable challenges for such approaches (*Ortiz et al., 2021*; *Venturelli et al., 2018*). Moreover, community dynamics observed in vivo can be difficult to reproduce under laboratory conditions, as the host presents a highly specialized

**eLife digest** Microbes colonize nearly every environment on Earth, from the ocean and soil to the inner and outer surfaces of animals, such as the gut or skin. They form communities that are usually made up of a diverse range of bacteria, often containing closely related species – a key factor for a successful community.

But closely related bacteria can battle for the same resources, so it is unclear how they manage to live alongside each other without competing against one another. While diet is thought to play a key role in enabling closely related bacterial species to co-exist in the gut of an animal, experimental evidence is lacking, due to the difficulty in replicating these systems in the laboratory.

One strategy for investigating microbial communities is using honeybees. A major dietary source for honeybees is pollen, which can also be applied in the laboratory to grow diverse types of bacteria found in the honeybee gut. In addition, scientists can generate bees that lack microbial communities in the gut, allowing them to add specific types of bacteria to study their impact.

Brochet et al. used this approach with Western honeybees to assess whether diet enables closely related bacteria to live alongside one another in the gut. First, they colonized bees that lacked gut microbes with four closely related bacteria of the genus *Lactobacillus*, alone or together, and fed the bees either sugar water or sugar water and pollen. After five days, the gut bacteria were analysed. This revealed that bees fed on sugar water only had one dominant *Lactobacillus* species present in their gut, while bees fed with additional pollen harboured all four *Lactobacillus* species. Further analysis of these four bacterial species revealed that each of them activates distinct genes when grown on pollen, allowing the different species to consume specific nutrients from broken down pollen.

These findings show that closely related bacteria can coexist in the gut by sharing the different nutrients provided in the diet of the host. Consequently, differences in dietary intake in honeybees and other animals may affect the diversity of gut bacteria, and potentially the health of an animal.

nutritional and spatial environment. Thus, there is a need for in vitro models that can reproduce ecological interactions observed in vivo, from simple co-culturing setups (*Li et al., 2019*) to sophisticated 'organoids-on-a-chip' systems (*Jalili-Firoozinezhad et al., 2019*; *Nikolaev et al., 2020*).

The gut microbiota of the Western honey bee (*Apis mellifera*) is composed of a few deep-branching phylogenetic lineages (phylotypes) belonging to the Firmicutes, Actinobacteria, and Proteobacteria phyla (*Martinson et al., 2011*; *Kwong and Moran, 2016*). Most of these lineages are composed of several closely related sequence-discrete populations, hereafter referred to as species, each of which contains further diversity at the strain-level (*Ellegaard and Engel, 2016*; *Ellegaard and Engel, 2019*; *Engel et al., 2012*; *Ellegaard et al., 2015*). Microbiota-depleted bees can be generated and experimentally colonized with synthetic communities of different strains. Moreover, most community members can be cultured in pollen, which is the major dietary source of honey bees (*Kešnerová et al., 2017*). This experimental tractability offers an excellent opportunity to probe the coexistence of bacteria in the gut of their native host and in controlled laboratory cultures using similar nutritional conditions.

One of the most abundant and diverse phylotype of the honey bee gut microbiota is *Lactobacillus* Firm5 (*Ellegaard and Engel, 2019*). This phylotype consists of facultative anaerobes that ferment sugars into organic acids and utilize various pollen-derived glycosylated plant compounds, such as flavonoids (*Kešnerová et al., 2017*). *Lactobacillus* Firm5 is specific to social bees but has diverged into many different species of which four are specifically associated with the Western honey bee, *Apis mellifera*: *Lactobacillus apis* (Lapi), *Lactobacillus helsingborgensis* (Lhel), *Lactobacillus melliventris* (Lmel), and *Lactobacillus kullabergensis* (Lkul). The four species are consistently present in the gut of individual honey bees suggesting that they can share the available niches and stably coexist despite their phylogenetic relatedness. Genomic analysis has revealed that these species share <85% pairwise average nucleotide identities (gANI) and exhibit high levels of genomic variation in terms of carbohydrate metabolism (*Ellegaard and Engel, 2019*; *Ellegaard et al., 2015*). However, whether the coexistence is facilitated by adaptation to different nutritional niches, and to what

extent the host environment, the diet, or the interactions with other community members matter is currently unknown.

Here, we tested under which conditions the four *Lactobacillus* Firm5 species can coexist and investigated the underlying molecular mechanism. We serially passaged the four species in vivo through gnotobiotic bees and in vitro in liquid cultures, and applied RNA sequencing and metabolomics analysis. Our results show that the coexistence of the four species is mediated by the partitioning of nutrients derived from the pollen diet of bees and is largely independent from the presence of the host or other community members.

## Results

### The coexistence of four related *Lactobacillus* species in the honey bee gut depends on the host diet

All experiments in this study were conducted with four bacterial isolates representing the four *Lactobacillus* Firm5 species (Lapi, Lhel, Lmel, and Lkul) associated with the Western honey bee. We first tested if the four species can establish in the gut of gnotobiotic bees under two different dietary conditions. To this end, we colonized microbiota-depleted bees with each of the four species, alone or together, and fed bees either sterilized sugar water (SW) or sterilized sugar water and pollen (SW+PG). Five days post-colonization, we assessed the bacterial loads in the gut by counting CFUs (*Figure 1A*, *Supplementary file 3*). When mono-colonized, the four species established in the gut of microbiota-depleted bees independent of the dietary treatment (*Figure 1A*). In the SW treatment, the colonization levels were generally lower than in the SW+PG treatment (*Figure 1A*, ANOVA q-value < 0.01) confirming previous results that pollen increases the total bacterial load in the gut (*Kešnerová et al., 2020*). There was no statistically significant difference between the total bacterial loads of the mono-colonizations and the co-colonizations in either dietary treatment, with the exception of the mono-colonization with Lkul, which showed higher loads than the co-colonizations in SW (*Figure 1A*, ANOVA q-value < 0.01). Consequently, the sum of the bacterial loads of the mono-colonizations exceeded the total bacterial load of the co-colonizations in both dietary treatments, suggesting that the species engage in negative interactions when colonizing the honey bee gut together.

To test if the four species can stably coexist in the bee gut, we serially passaged the community seven times in microbiota-depleted bees under both dietary conditions (SW and SW+PG). After each passage (i.e. after 5 days of colonization), we used amplicon sequencing of a

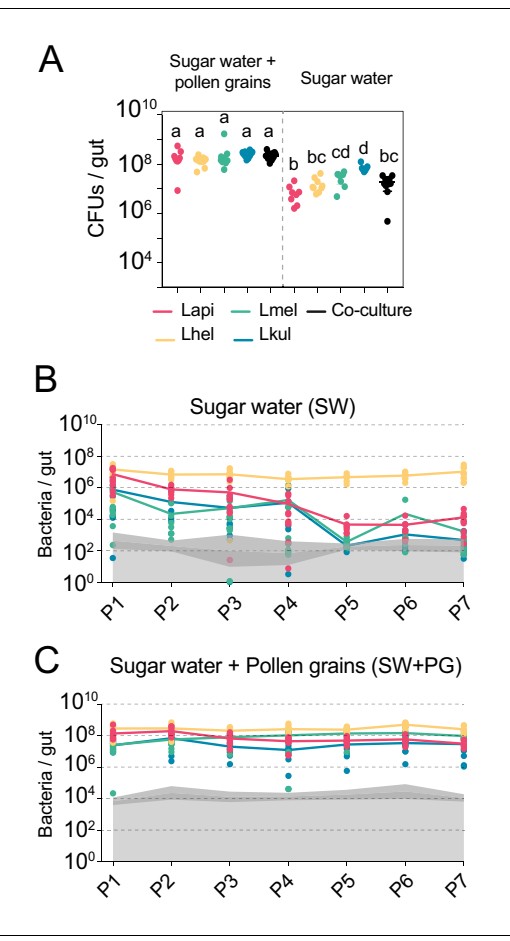

**Figure 1.** The presence of dietary pollen facilitates the stable coexistence of the four *Lactobacillus* species in the honey bee gut. (**A**) Bacterial abundance (CFUs) in the gut of gnotobiotic bees (n = 7–10) colonized with the four species separately or together under two different dietary conditions. Bees were sampled five days after colonization. Statistical differences (ANOVA with Tuckey post-hoc test and BH correction) are depicted by different letters. (**B, C**) Changes in the absolute abundance of each member of the four-species community across the seven serial passages (P1–P7) through the gut of gnotobiotic bees. The absolute abundance of each species was determined by multiplying the total number of CFUs with the relative abundance of each species in the community. Grey areas represent the limit of detection which can vary depending on the sequencing depth of each replicate (see Materials and methods). Therefore, the average limit of detection and the 95% confidence intervals are shown.

discriminatory housekeeping gene fragment (see Materials and methods) in combination with CFU counting to determine the absolute abundance of each species in the community. We observed clear differences between the two dietary conditions in the ability of the four species to coexist across the passages (*Figure 1B–C*, *Supplementary file 4*). In the SW treatment, all species were initially detectable in most samples (P1, *Figure 1B*). However, three species (Lapi, Lmel and Lkul) steadily decreased in abundance in the subsequent passages resulting in a rapid dominance of Lhel (*Figure 1B*). Lmel and Lkul reached the detection limit and Lapi decreased to around $10^4$ bacteria/gut by passage five (P5, *Figure 1B*). Only Lhel was stably maintained across all seven passages and was present at around 1000x higher abundance than Lapi at the end of the experiment (~$10^7$ bacteria/gut, *Figure 1B*). In the contrary, in the SW+PG treatment, all four species were detectable in all passages at around $10^6$ to $10^8$ bacteria/gut, and displayed a highly stable abundance profile over time (*Figure 1C*).

In summary, these findings show that the four species can stably coexist in vivo when bees are fed pollen, but not when they are only fed sugar water. This is consistent with the idea that pollen facilitates niche partitioning in the honey bee gut by offering competing species different ecological niches facilitating their coexistence.

## In vitro co-culture experiments recapitulate the nutrient-dependent coexistence of the four *Lactobacillus* species

We next tested if the four species can also coexist in vitro, outside of the host environment, under different nutrient conditions. To this end, we cultured the species alone or together in minimal medium supplemented with either glucose (G), pollen extract (PE), or entire pollen grains (PG). All four species were able to grow when cultured alone under the three nutrient conditions (*Figure 2—figure supplement 1*, *Supplementary file 3*). Growth yields of Lhel, Lkul, and the co-culture were slightly lower in PE and PG than in G, and Lmel showed lower growth yields than some of the other species in PE and G (*Figure 2—figure supplement 1*, ANOVA q-value < 0.01). As in vivo, the total bacterial loads of the co-cultures were not consistently different from those of the mono-cultures (*Figure 2—figure supplement 1*), suggesting that the four species have overlapping metabolic niches and engage in negative interactions with each other.

We then serially passaged the co-cultures 21 times under the three different nutrient conditions by transferring an aliquot after 24 hr of growth into fresh culture medium (1:20). The absolute abundance of each strain was determined after every other passage by combining amplicon sequencing with qPCR (see Materials and methods). As for the in vivo experiment, we observed clear differences in the growth dynamics of the four species, both over time and between the glucose and the pollen culture conditions (*Figure 2*, *Supplementary file 4*). In the presence of glucose, three of the four species (Lhel, Lmel, and Lkul) steadily decreased in abundance over time (*Figure 2A*), with two of them reaching the limit of detection (<$10^5$ bacteria/ml) after about 11 passages (P11). In contrast, Lapi was stably maintained at high abundance ($10^9$ bacteria/ml) until the last passage (*Figure 2A*) and hence dominated the co-culture for most of the transfer experiment. In the presence of PE or PG, the four species revealed very different growth behaviors (*Figure 2B and C*). None of the species decreased over time, and after 21 transfers all species still yielded between $10^6$ and $10^9$ bacteria/ml.

To look at changes in community composition over time, we measured the community stability (temporal mean divided by temporal standard deviation of the species abundances) in sliding windows of five passages. Little to no change in community stability was observed for the two pollen conditions throughout the experiment, whereas in glucose the community reached a stable state after ~11 transfers (*Figure 2D*). To compare the growth yields of each species across the three nutrient conditions, we only considered the passages after which community stability was reached (P13-21). With the exception of Lapi all species reached higher yields in the presence of pollen as compared to glucose (*Figure 2E*, ANOVA q-value < 0.01). Notably, Lmel was the only species that showed improved growth in PG as compared to PE (*Figure 2A–C*).

In summary, these findings show that the nutrient-dependent coexistence of the four *Lactobacillus* species observed in vivo can be recapitulated in vitro in a simple co-culture experiment, suggesting that the partitioning of pollen-derived nutrients is sufficient for enabling coexistence. Similar results were obtained for a second in vitro experiment which included the same nutrient conditions, but was only conducted for ten transfers (*Figure 2—figure supplement 2*).

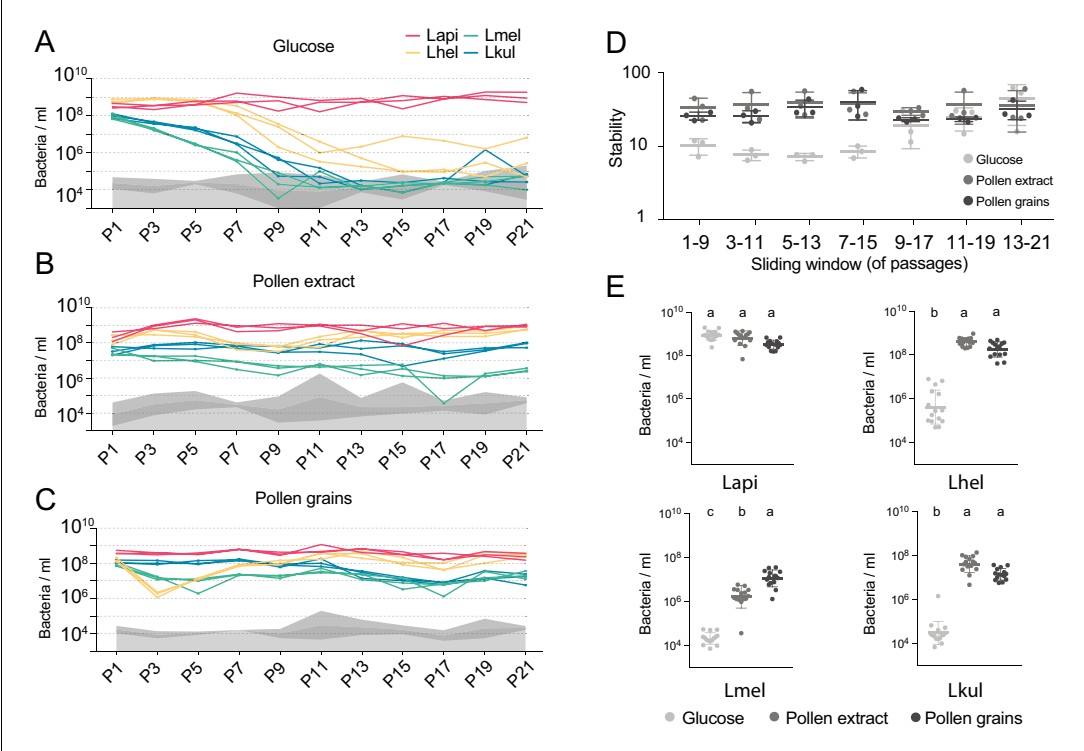

**Figure 2.** The stable coexistence of the four *Lactobacillus* species can be recapitulated in vitro in the presence of pollen. (A–C) Changes in total abundance of the four species when serial passaged in co-culture for 21 times in minimal medium supplemented with (A) 2% (w/v) glucose, (B) 10% (v/v) pollen extract, and (C) 10% (v/v) pollen grains. The absolute abundance of each species was determined by multiplying the total number of CFUs with the proportion of each strain in a given sample as based on amplicon sequencing (see Materials and methods). Gray areas indicate the limit of detection as explained in the Materials and methods. (D) Community stability of each replicate calculated based on the species abundances for a sliding window of five passages with a step size of 1. (E) Absolute abundance of each species across the three treatments considering the replicates of passages 13–21, which is when the community reached stability. Statistical differences (ANOVA with Tuckey post-hoc test and BH correction) are depicted by different letters.

The online version of this article includes the following figure supplement(s) for figure 2:

**Figure supplement 1.** Colony-forming units (CFUs) per ml of culture after 24 hr of growth of the four species in mono-cultures (n=3) or in co-culture (n=3) in the presence of 2% (w/v) glucose (G), 10% pollen extract (PE), or 10% pollen grains (PG).

**Figure supplement 2.** Second in vitro transfer experiment.

## The four *Lactobacillus* species upregulate divergent carbohydrate transport and metabolism functions in the presence of pollen during gut colonization

Given the impact of pollen on the coexistence of the four *Lactobacillus* species, we tested if genes involved in nutrient acquisition and metabolism were differentially expressed between the dietary treatments. To this end, we carried out RNA sequencing of the four-species community in honey bees that were fed either sugar water (SW) or sugar water and pollen grains (SW+PG) (*Figure 3A*). Multidimensional scaling (MDS) of the normalized read counts mapped to each species revealed that most samples clustered by treatment (SW+PG versus SW) (*Figure 3—figure supplement 1*), indicating that all four species exhibited dietary-specific transcriptional responses.

We found a total 687 genes (181 to 217 genes per species) to be differentially expressed ($\log_2$FC ≥ |2| and p-value ≤ 0.01) between the two dietary treatments (*Figure 3B*). 'Carbohydrate transport and metabolism' (Cluster of orthologous group category G, COG G) was by far the most abundant functional category among the genes upregulated in the SW+PG treatment relative to the SW treatment (*Figure 3C*, 17.1–37.6% of all upregulated genes). In three of the four species (Lmel, Lhel, and Lkul), this category was significantly enriched among the upregulated genes (Fisher's exact test, p<0.01, *Supplementary file 6*). The largest fraction of the upregulated COG G genes encoded PTS

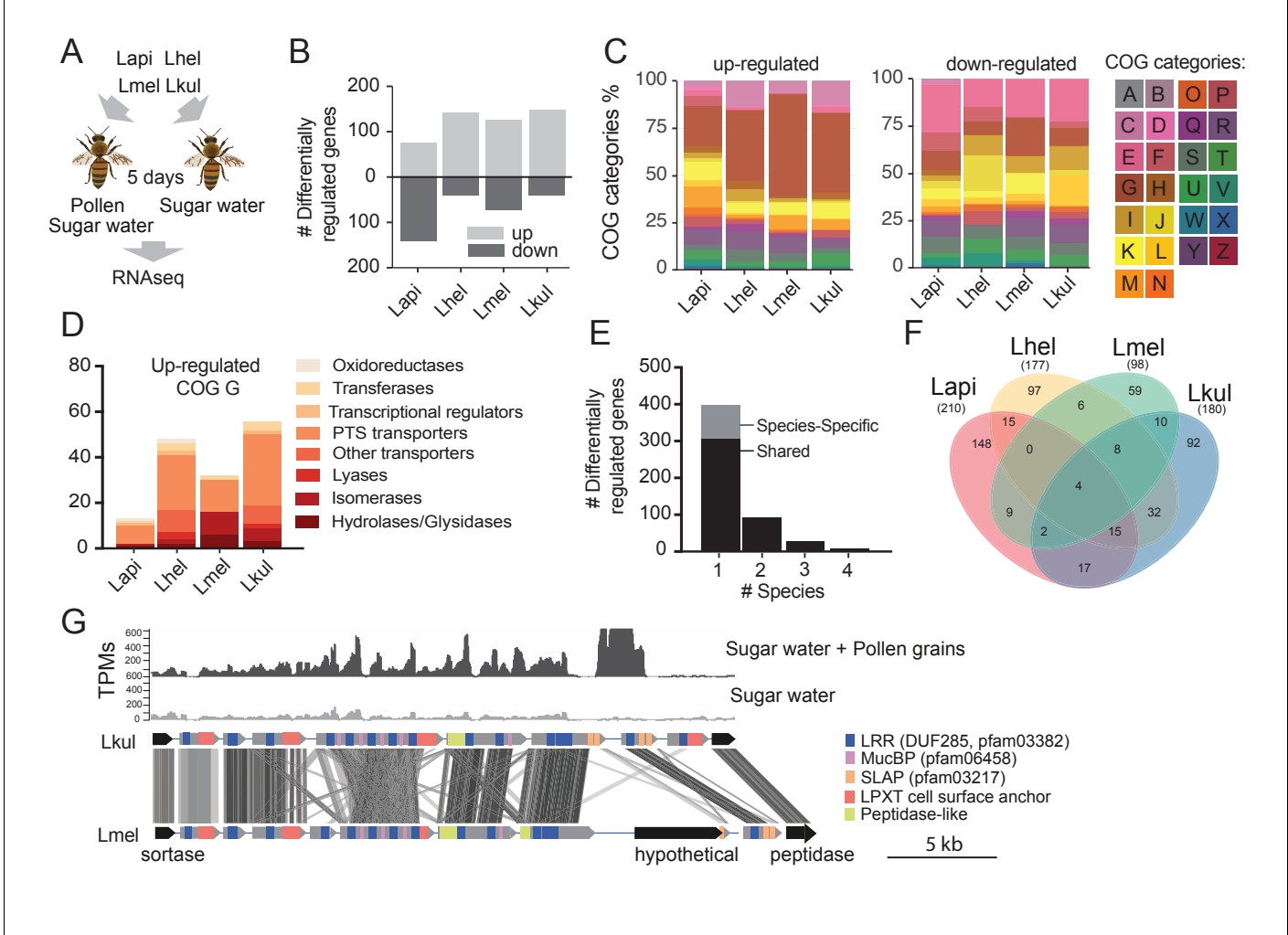

**Figure 3.** Transcriptome analysis of the four *Lactobacillus* species during co-colonization of gnotobiotic bees. (A) Schematic outline of the RNA-Seq experiment. (B) Number of differentially regulated genes (log2FC ≥ |2| and p-value ≤ 0.01) in each species during co-colonization of gnotobiotic bees fed either pollen and sugar water (PG+SW) or sugar water only (SW). Up- and down-regulated genes are shown in different gray tones. (C) COG categories of genes up- or down-regulated by the four species in SW+PG if compared to SW. For COG definitions, see ***Supplementary file 2*** (D) Functional sub-categories of COG 'G' genes upregulated in SW+PG if compared to SW. (E) Barplot displaying numbers of gene families differentially regulated in one species, two species, three species, or four species. Gene families differentially regulated in only one species are split into those that have homologs in the other species or that are species-specific.(F) Venn diagram showing overlap of gene families (based on gene homology) differentially regulated in the four species. (G) Transcripts per million (TPM) for two representative samples of the SW+PG and the SW treatments over a genomic region of Lkul encoding *Lactobacillus*-specific surface proteins. The genomic region of Lkul is compared to a similar region identified in Lmel which is also differentially regulated across the two treatments (expression profile not shown). Similarity between genes is shown by vertical lines. Gray tones indicate level of similarity. Surface protein-encoding genes are show in grey with the different domains and motifs shown according to the color legend.

The online version of this article includes the following figure supplement(s) for figure 3:

**Figure supplement 1.** MDS plots of in vivo RNA-seq samples.

transporters (***Figure 3D***, ***Supplementary file 5***), followed by other sugar transporters (e.g. ABC transporters), and enzymes involved in sugar cleavage and conversion (***Figure 3D***). Among the downregulated genes, COG G genes were not abundant (5.1–7.8%) (***Figure 3C***). Instead, the category 'Amino acid metabolism and transport' (COG E) was enriched in Lapi (Fisher's exact test, p < 0.01, ***Supplementary file 8***), and genes encoding ABC-type amino acid transporters were present among the downregulated genes in all species (***Supplementary file 5***).

We next clustered all genes by homology into gene families. While most of the differentially expressed genes (89%) belonged to gene families with homologs in multiple species, differential expression was typically observed for just one of the species (*Figure 3E–F*). This suggests that the presence of pollen triggers distinct transcriptional changes in the four species during gut colonization. Indeed, gene annotation analysis allowed us to identify several species-specific metabolic functions among the differentially regulated genes (*Figure 4*, *Supplementary file 5*). For example, Lhel specifically upregulated three PTS gene clusters for the uptake and metabolism of sugar alcohols and one gene cluster for ribose utilization. In contrast, Lmel upregulated several gene clusters involved in the cleavage of xylose, mannose, rhamnose, and arabinose from polysaccharides or other glycosylated compounds. Lmel also upregulated a gene cluster for the synthesis and the transport of bacteriocins in the presence of pollen. Lkul upregulated a starch utilization gene cluster, which in part was also differentially regulated in Lmel. In addition, this species upregulated an oligopeptide transporter gene cluster that was present in some of the other strains but not differentially regulated. The fourth species, Lapi, also differentially expressed genes belonging to COG 'G' (mainly PTS transporters), but fewer ones, and with similar functional annotations as found in the other three species. However, Lapi was the only species that upregulated two conserved deoxycytidine kinase genes encoding enzymes involved in nucleoside salvage pathways.

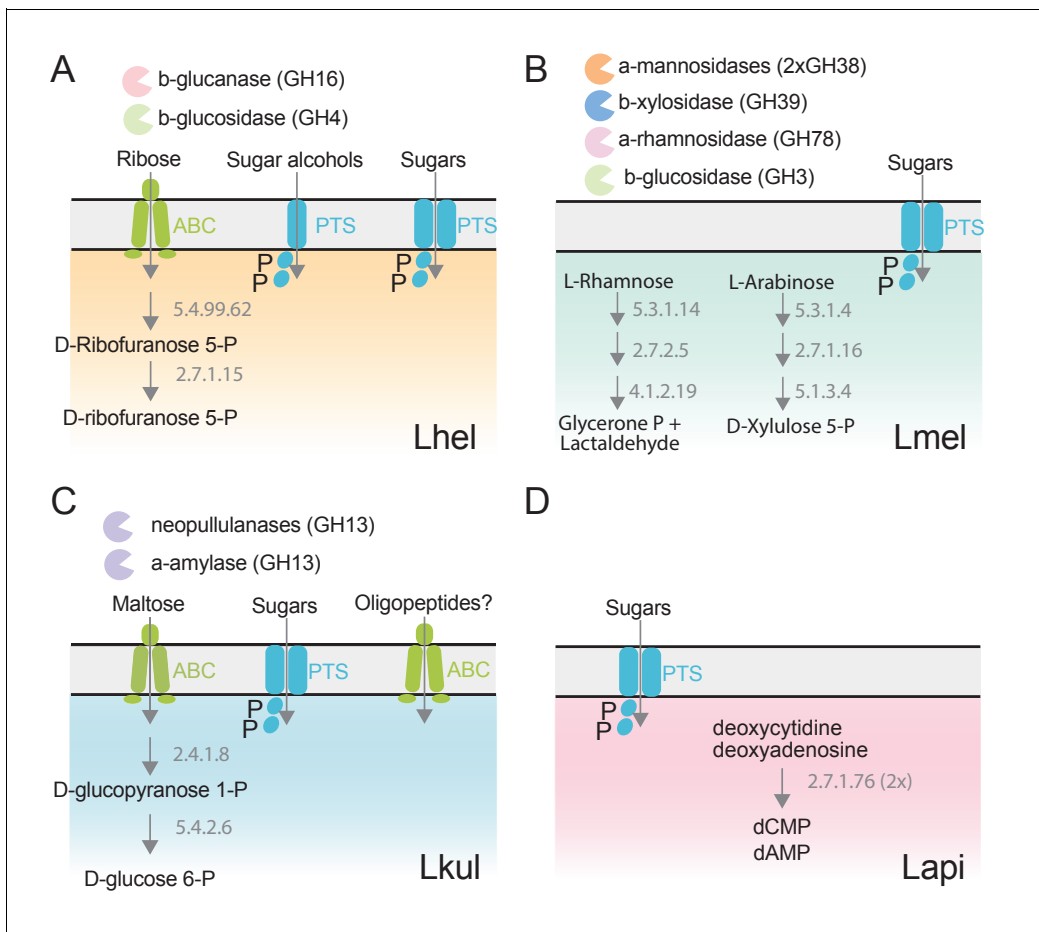

**Figure 4.** The four *Lactobacillus* species upregulate different carbohydrate metabolism functions during gut colonization of gnotobiotic bees. (**A**) Lhel, (**B**) Lmel, (**C**) Lkul, (**D**) Lapi. Only enzymes and transporters that are upregulated in a species-specific manner in the pollen treatment versus the sugar water treatment are shown. The figure is not exhaustive, but highlights the main differences that could be identified based on gene annotations among all differentially regulated genes (*Supplementary file 6*). Glycosidases belonging to different CAZyme families are represented by different colors. ABC: ABC transporters, PTS: phosphotransferase system transporters. Numbers indicate EC numbers of upregulated enzymatic steps.

Besides these species-specific transcriptional changes, a number of interesting functions were differentially regulated in more than one species. For example, we found evidence for citrate fermentation in Lhel and Lkul. Both species upregulated genes encoding a citrate lyase for the conversion of citrate into oxaloacetate and acetate in the presence of pollen (*Supplementary file 5*). Lhel, Lmel, and Lkul upregulated genes for the uptake and metabolism of glycerol. Moreover, all four species upregulated gene clusters encoding surface proteins with leucine-rich repeat (LRR) regions, LPXT cell-wall anchoring motifs, and SLAP (S-layer associated protein) domains (*Figure 3G*).

Altogether, these results suggest that the four species utilize different carbohydrate-related resources from pollen, which supports the niche partitioning hypothesis as the basis for coexistence.

## Transcriptional responses to pollen are similar in vivo and in vitro

In vivo gene expression differences between the two dietary conditions could be influenced by the host or by bacteria-bacteria interactions. Therefore, we carried out an additional transcriptomics analysis to disentangle the contribution of each of these factors to transcriptional changes in the four *Lactobacillus* species. We grew the four species in vitro in either co-culture or mono-culture, and with either pollen extract (PE) or glucose as growth substrate (G) (*Figure 5A*). As for the in vivo RNA-Seq analysis, MDS plots of the normalized read counts indicated that the four species exhibit treatment-specific transcriptional responses (*Figure 5—figure supplement 1*).

For each species, whether grown alone or in co-culture, we found between 159 and 393 genes to be differentially regulated between the PE and the G treatment (*Figure 5B*, $\log_2FC \geq |2|$ and p-value$\leq$0.01). As in vivo, Carbohydrate transport and metabolism (COG 'G') was the predominant functional category among the upregulated genes in the presence of pollen (*Figure 5C*) and enriched in all eight comparisons (four species, each alone or in co-culture, Fisher's exact test p-value < 0.01, *Supplementary file 8*). Moreover, 25.3–36.9% of the genes upregulated in vivo were also upregulated in vitro in the presence of pollen. In particular, the species-specific carbohydrate metabolism functions described above (*Figure 4*) showed a similar transcriptional response to pollen in vivo and in vitro (*Figure 5D*). In contrast, most of the putative adhesin genes upregulated in vivo were not upregulated in vitro during growth in pollen or had relatively low transcripts per million (TPM). This suggests that these genes are either expressed in response to the host environment, or the presence of entire pollen grains or sugar water, both of which were only included in the in vivo but not in the in vitro experiment (*Supplementary file 9*). It is also noteworthy that fewer genes were downregulated than upregulated in pollen relative to glucose, and that the COG category 'G' was not enriched among the downregulated genes, which is concordant with our in vivo transcriptome analysis. (*Supplementary file 8*). Based on these results, we conclude that each species upregulates specific operons for the transport and utilization of different carbohydrates (e.g. sugar alcohols and glycans) in response to the presence of pollen, independent of the host environment.

## The presence of other community members has little impact on the transcriptional profile of the four species

We found that a large fraction of the genes upregulated in PE relative to G in the mono-cultures were also upregulated in the co-cultures (58.2–87.8%, *Figure 5E*). In particular, the gene clusters identified to be regulated in a species-specific manner (see above) showed highly concordant gene expression profiles in vitro independent of the presence/absence of the other *Lactobacillus* species. This was confirmed by the direct comparison of mono-culture and co-culture conditions. In comparison to the nutritional treatments, fewer genes (9–149 genes) were differentially expressed between co-culture and mono-culture treatments ($\log_2FC \geq |2|$ and p-value$\leq$0.01), (*Figure 5F*).

We could not find any consistent pattern across the four species in terms of COG category enrichment (*Supplementary file 8*). Moreover, only a few genes were differentially expressed in more than one species (6.25–30%), or across both nutrient conditions (1.86–5.33%). Citrate fermentation genes were upregulated in Lkul in co-culture relative to mono-culture when grown in pollen, whereas in Lhel the opposite was observed (*Figure 5G*). Also of note, the oligopeptide transporter system which was upregulated in vivo in Lkul in the presence of pollen, was also upregulated in vitro in the presence of pollen, but only when other species were present. These two specific examples show that a few metabolic functions are differentially regulated in response to other bacteria, but not always in the same direction across species, or only in a specific nutrient condition. We thus

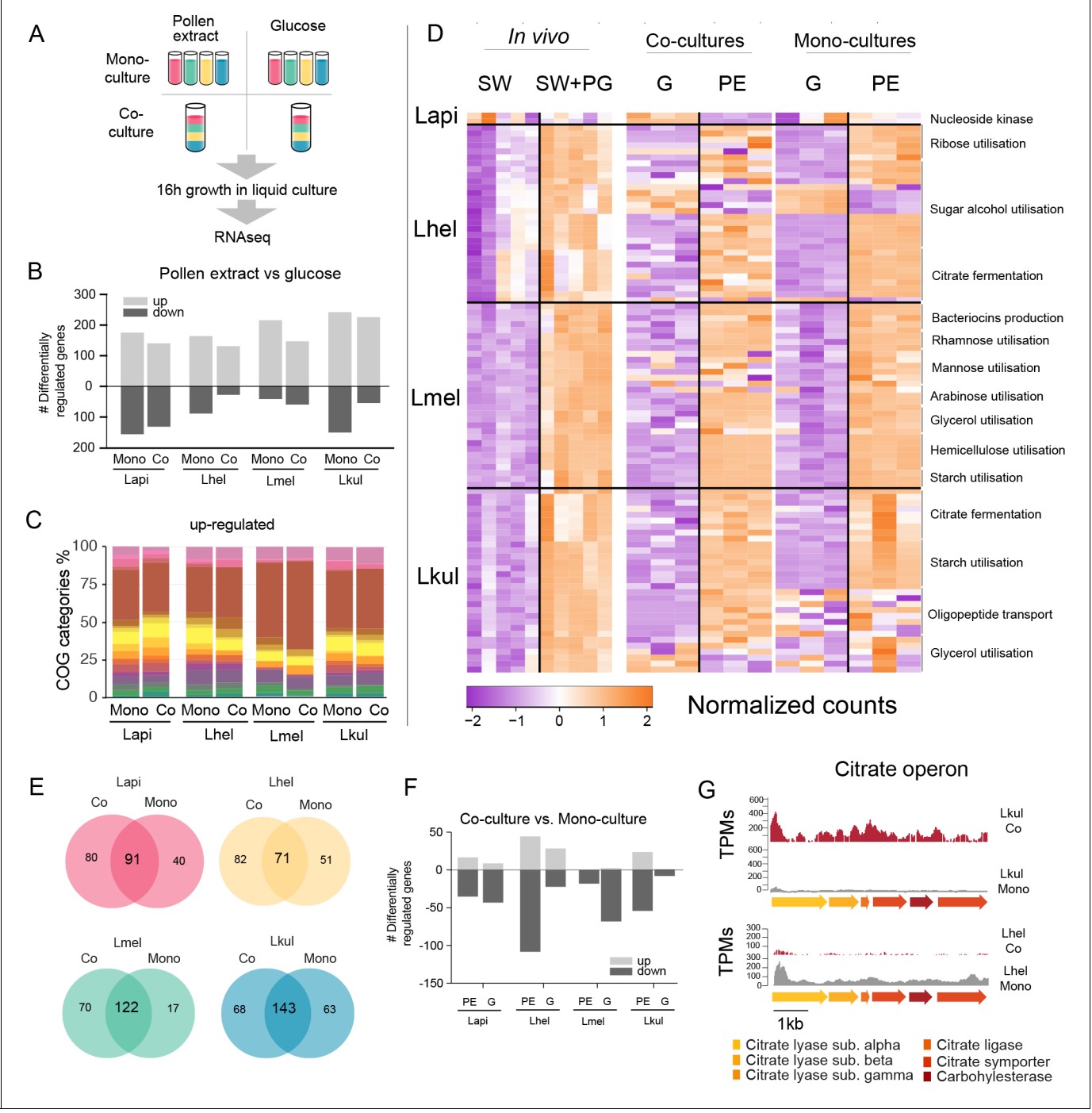

**Figure 5.** Transcriptome analysis of the four *Lactobacillus* species grown in vitro in pollen extract or in glucose. (A) Scheme of the 2x2 experimental design. Species were grown alone or together, in either glucose (**G**) or pollen extract (PE). (B) Number of differentially regulated genes in each of the four species in the presence of PE if compared to G. Mono, mono-culture, Co, co-culture. Up- and down-regulated genes are shown in different gray tones. (C) COG categories of genes up-regulated by the four species in the presence of PE if compared to G. The colors are the same as in *Figure 3C*. For COG definitions, see *Supplementary file 2* Heatmap displaying normalized counts of selected genes differentially regulated across the in vivo and in vitro RNA-Seq experiments. We selected metabolic genes and gene clusters that were identified in the in vivo experiment to be differentially regulated across the two treatments and which could be assigned a putative function based on annotation. Counts were normalized for each gene and dataset separately, that is in vivo, co-cultures, and mono-cultures. (E) Venn diagrams displaying the overlap of the genes differentially regulated between the PE and G treatment when the four species were grown in co-culture (Co) and mono-culture (Mono). (F) Number of differentially regulated genes in each of the four species in co-culture relative to mono-culture. Up- and down-regulated genes are shown in different gray tones. (G)

*Figure 5 continued on next page*

*Figure 5 continued*

Transcripts per million (TPM) over a genomic region of Lkul and Lmel encoding genes for citrate fermentation (i.e. citrate operon) for a representative sample of the co-culture and the mono-culture treatment when grown in PE.

The online version of this article includes the following figure supplement(s) for figure 5:

**Figure supplement 1.** MDS plots of in vitro RNA-seq samples.

conclude that the main factor driving changes in gene expression in the four strains is the presence of pollen, rather than the presence of other *Lactobacillus* species.

## Metabolomics analysis reveals differences in flavonoid and sugar metabolism across the four *Lactobacillus* species

Our transcriptome analyses suggest that differences in sugar metabolism may enable the four species to coexist in the presence of pollen in vitro and in vivo. To assess species-specific metabolic changes when grown in pollen, we profiled the metabolome of the pollen extract medium before (t = 0 hr) and after bacterial growth (t = 16 hr) using Q-TOF-based untargeted metabolomics (*Fuhrer et al., 2011*). We annotated a total of 657 ions of which 406 could be reliably categorized as pollen-derived ions, as opposed to ions originating from the base medium (see Materials and methods, *Supplementary file 10*, *Figure 6—figure supplement 2*). The metabolomics data clearly separated the four species indicating distinctive metabolic changes and thus corroborating the transcriptome results (*Figure 6—figure supplement 1*). A total of 76 pollen-derived ions showed a significant decrease in abundance over-time (log2FC $\leq -1$ and p-value$\leq$0.01, Student's t-Test, BH correction) (*Figure 6A*, *Supplementary file 10*). Of those, 24 ions decreased in abundance in all four species, another 24 ions decreased in abundance in only a subset of the species, and the remaining 28 ions decreased in abundance in only a single species (*Figure 6A*). Ions annotated as glycosylated flavonoids were among the top ions responsible for the separation of the four species in the PCA (*Figure 6—figure supplement 1*). Lmel depleted six different ions annotated as flavonoids (isoorientin 2''-O-rhamnoside, quercetin-3-O-glucoside, vitexin, rutin, luteolin-7-O-(6''-malonylglucoside), and quercetin-3-O-beta-D-glucosyl-(1->2)-beta-D-glucoside), while Lapi depleted three ions annotated as flavonoids (isoorientin 2''-O-rhamnoside, quercetin-3-O-glucoside, vitexin) (*Figure 6A*, *Figure 6—figure supplement 3*). In contrast, Lkul only depleted the flavonoid ion annotated as isoorientin 2''-O-rhamnoside, and no flavonoid ion changes were identified for Lhel (*Figure 6A*, *Figure 6—figure supplement 3*).

To corroborate the species-specific utilization of flavonoids, we incubated each of the four species in base culture medium supplemented with rutin. We observed the formation of a yellow insoluble precipitate only in the wells incubated with Lmel (*Figure 6B*). Metabolomics analysis confirmed that rutin was depleted in these wells and that the yellow precipitate corresponded to an accumulation of quercetin, the water-insoluble, deglycosylated aglycone of rutin (*Figure 6C*). These findings are consistent with our transcriptome results which show that Lmel is the only species that upregulated a rhamnosidase gene known to cleave rhamnose residue from rutin (*Beekwilder et al., 2009*; *Figure 4*).

Other ions with species-specific abundance changes included a plant-derived glycosylated compound belonging to the iridioids class (i.e. antirrhinoside, depleted in the presence of Lapi), a component of the outer pollen wall (i.e. 9,10,18-trihydroxystearate, accumulated in the presence of Lmel) and cyclic nucleotides (depleted in the presence of Lhel, Lmel, and Lkul) (*Figure 6A* and *Figure 6—figure supplement 3*). Lhel was the only species depleting an ion corresponding to sugar alcohols (mannitol, D-sorbitol, or L-glucitol) (*Figure 6A* and *Figure 6—figure supplement 3*) consistent with the specific upregulation of sugar alcohol PTS transporters in this species (*Figure 4*).

Based on the untargeted metabolomics analysis, we conclude that the four species target different metabolites, in particular secondary plant metabolites present in pollen. In order to assess differences in the utilization of simple sugars and acids in more detail, we analyzed the supernatants of cultures of the four strains after 0, 8, 16, and 24 hr of growth using GC-MS. We used a semi-targeted approach, where we identified a subset of metabolites by preparing analytical standards and the others by using a reference library (see Materials and methods). We identified 113 metabolites of which 46 showed a significant change in abundance in at least one strain between timepoint 0 hr

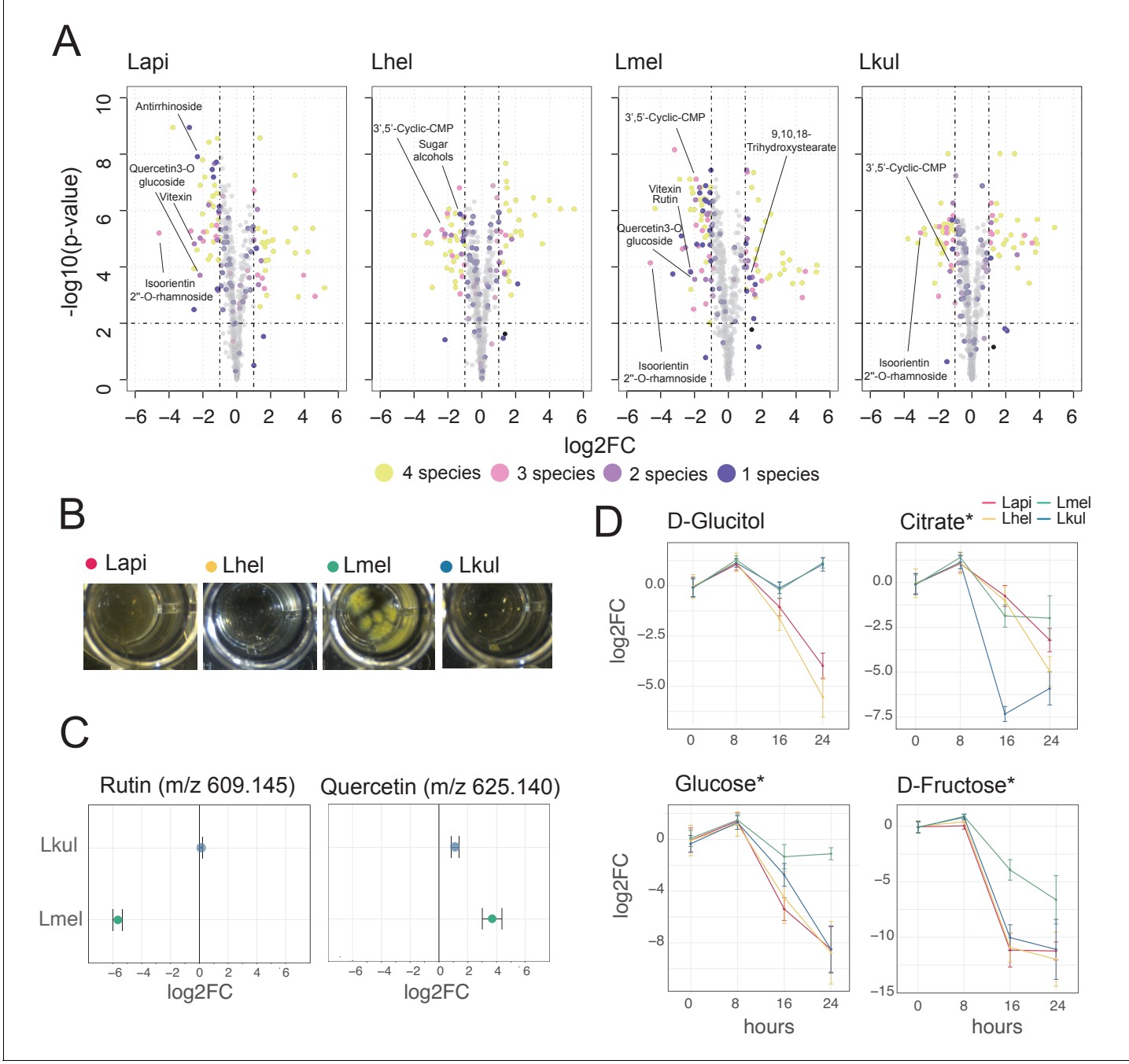

**Figure 6.** Metabolomics analysis shows differences in the utilization of pollen-derived glycosides across the four *Lactobacillus* species. (A) Volcano plots displaying ions with significant fold changes (FC) for each of the four species after 16 hr of growth in pollen extract versus glucose. Each dot corresponds to an ion in the untargeted metabolomics dataset. Different colors represent ions that significantly change over time in one, two, three, or four species. Dashed black lines represent the significance thresholds: p-value < 0.01 and log2FC < −1 or > 1. (B) Culture wells of the four species grown in cfMRS + 0.05% rutin after 16 hr of incubation. The yellow precipitate is only visible for Lmel. (C) Rutin and quercetin detection in spent medium of Lmel and Lkul grown in cfMRS + 0.05% rutin after 16 hr of incubation (n=5). (D) Changes in key metabolites during growth measured by GC-MS (n=5). Log2FC relative to time point 0 is plotted. Time is reported in hours. * Indicates metabolites whose identity was confirmed using analytical standards. For m/z values see *Supplementary file 10*.

The online version of this article includes the following figure supplement(s) for figure 6:

**Figure supplement 1.** PCA in vitro metabolomics.

**Figure supplement 2.** Definition of pollen-derived ions.

**Figure supplement 3.** Untargeted metabolomics: key metabolites discussed in the main text.

*Figure 6 continued on next page*

*Figure 6 continued*

**Figure supplement 4.** GC-MS detection of key metabolites over time.
**Figure supplement 5.** Logistic regression growth curve of the four species.

and 24 hr (log$_2$FC $\geq$ |2| and p-value $\leq$ 0.01, Student's t-Test, BH correction) (**Supplementary file 10**). All four species showed mixed substrate utilization, that is they utilized several substrates simultaneously. Moreover, most substrates were utilized by all four species, but often at different rates. Many metabolites that we identified with the GC-MS had annotations comparable to the ones found in the Q-TOF-based experiment. For example, we detected a compound annotated as sugar alcohol, that is glucitol, that was consumed most efficiently by Lhel as observed in the previous analysis (**Figure 6C**). Moreover, all four species consumed the carboxylic acids citrate and malate (**Figure 6C** and **Figure 6—figure supplement 4**), which corresponded with the results of the Q-TOF-based experiment. Interestingly, Lkul and Lhel consumed citrate at the fastest rate and they were also the two species that upregulated gene clusters for citrate fermentation in the presence of pollen in vivo and in vitro (**Figure 4**).

Lmel consumed several simple monosaccharides (such as glucose, fructose, allose, and mannose) at a slower rate than the other species, although having a similar growth profile (**Figure 6C**, **Figure 6—figure supplements 4–5**). This could indicate that Lmel has specialized in the metabolism of pollen-derived glycosylated compounds (such as rutin, **Figure 6B–C**) at the expense of fast consumption of generic substrates, which accords with the upregulation of several gene clusters for the cleavage of such sugars from polysaccharides or other glycosides (e.g. flavonoids) in presence of pollen (**Figure 4**).

In summary, our metabolomics results show that the four species specialize in the utilization of different pollen-derived compounds, and that the observed metabolite changes are to some extent consistent with the transcriptional changes observed in the presence of pollen relative to the presence of simple sugars.

## Discussion

Ecological processes governing the coexistence of microbes have been probed in the laboratory using microbial communities of different complexity (**Goldford et al., 2018**; **Ortiz et al., 2021**; **Wright and Vetsigian, 2016**; **Friedman et al., 2017**; **Piccardi et al., 2019**; **Deines et al., 2020**; **Logan, 2017**). However, few studies have examined the impact of the host on the coexistence of bacterial symbionts of animals (**Ortiz et al., 2021**; **Deines et al., 2020**). In particular, little is known about the extent to which closely related species and strains can be stably maintained (**Bittleston et al., 2019**). We capitalized on the experimental tractability of honey bees and their gut microbiota and used a bottom-up approach to study the coexistence of four closely related, naturally co-occurring *Lactobacillus* species. We disentangled the effect of the diet and the host on the interactions between the four species by serially passaging them through gnotobiotic bees or in culture tubes, under two nutrient conditions (pollen versus simple sugars). Our results show that the dynamics in the four-species community is governed by negative interactions, because the growth of each member was lower in co-culture than in mono-culture, independent of the environment (host or culture tube) and the nutrient condition (pollen or simple sugars). This is consistent with previous observations that negative interactions predominate in nutrient-rich environments (**Piccardi et al., 2019**; **Foster and Bell, 2012**; **Berry and Widder, 2014**; **Coyte et al., 2015**; **Ghoul and Mitri, 2016**). Moreover, the four *Lactobacillus* species harbor relatively small genomes (1.5–2 Mb) with a conserved and streamlined core metabolism and similar auxotrophies, suggesting overlapping nutritional requirements (**Ellegaard et al., 2019**; **Ellegaard et al., 2015**; **Kwong and Moran, 2016**).

The coexistence of bacterial symbionts can be facilitated by the host, for example by providing a spatially structured environment that results in the physical separation of competing strains (**Gude et al., 2020**; **Kim et al., 2008**; **Mitri et al., 2016**; **Hallatschek et al., 2007**), or by secreting metabolites that support niche specialization (**Schluter and Foster, 2012**; **McLoughlin et al., 2016**). However, in the case of the four *Lactobacillus* species, such host-related features seem not to be sufficient to support coexistence, because the four-species community was rapidly dominated by a

single species, when passaged through gnotobiotic bees that were fed a simple sugar diet. In contrast, when providing a more diverse nutrition in the form of pollen, we found that the four species were stably maintained both in vivo and in vitro. We thus conclude that the coexistence of the four *Lactobacillus* species in the honey bee gut primarily depends on the pollen diet of the host and not the host environment itself.

The challenges in replicating the native environment such that it is possible to study relevant interactions of host-associated microbes in vitro are formidable. These were highlighted in a recent study on the microbial community associated with the freshwater polyp hydra that could not recapitulate the coexistence of the dominant microbiota members in vitro (*Deines et al., 2020*). Here, we aimed to approximate the nutritional conditions in the honey bee gut by culturing the bacteria in pollen infused media, that is the natural diet of bees. In both the in vivo and in vitro transfer experiment, we assessed the effect of pollen on the dynamics of the community by comparing it to a simple sugar treatment. Although not identical, the nutritional conditions in vitro were sufficiently similar as to recapitulate the overall community dynamics observed in vivo: pollen nutrients supported the stable coexistence of the four species, while the simple sugars led to the dominance of a single species. As the bee and members of the bee gut microbiota pre-digest pollen and sugars upstream of the rectum, it is difficult to exactly replicate the metabolic environment of the rectum. For example, sucrose is largely absorbed via the midgut epithelium and cleaved into glucose and fructose by host enzymes, while fermentative bacteria such as *Gilliamella apicola* degrade and modify a diverse range of carbohydrates in the ileum (*Kešnerová et al., 2017*; *Crailsheim, 1988*). These metabolic alterations may explain some of the differences observed between the in vivo and in vitro experiments, such as the dominance of different species in the simple sugar conditions (sucrose and glucose, respectively). We therefore suspect that different species would dominate in vitro or in vivo with an alternative simple sugar composition.

Our findings are consistent with the consumer-resource model, which predicts that the number of species that can coexist depends on the number of available resources (*Tilman, 1986*). Correspondingly, in the presence of a single substrate, such as in the case of glucose in vitro, competition for the same nutrient results in the competitive exclusion of all but one species. However, depending on the nutrient availability, the dietary transit time, the crosstalk with the host, or the spatial structure of the gut, the ecological processes governing bacterial coexistence may differ across host-associated microbiomes. For example, the *Lactobacillus* species of the honey bee gut microbiota primarily colonize the luminal space of the rectum, where partially digested pollen accumulates. In contrast, some of the Proteobacteria of the bee gut microbiota adhere to the epithelial surface of the ileum (*Zheng et al., 2018*). We expect that in the latter case interactions with the host play a more important role for microbial coexistence than in the case of the Lactobacilli in the rectum.

Although ecological interactions in bacterial communities have been investigated across a wide range of experimental systems, few studies have tackled the molecular mechanisms underlying coexistence. In some cases, cross-feeding of metabolic by-products facilitates the maintenance of diversity in bacterial communities, such as after passaging leaf and soil samples in single carbon sources (*Goldford et al., 2018*). However, cross-feeding does not seem to play an important role in maintaining coexistence of the four *Lactobacillus* species in this study. Unlike the above example, feeding a single carbon source led to the extinction of all but one species. Our metabolomics analysis also did not reveal any major metabolites that could potentially be cross-fed, that is were produced by one species and utilized by another. Finally, we identified no transcriptional changes that would suggest cross-feeding activities when comparing mono-cultures and co-cultures of the four *Lactobacillus* species.

Instead, our combined transcriptomics and metabolomics analyses suggest that coexistence is facilitated by specialization toward distinct pollen-derived nutrients. We found that all four species upregulated carbohydrate transport and metabolism functions dedicated to the utilization of different carbon sources in the presence of pollen when colonizing the bee gut, and these changes were reproducible in vitro. Our metabolomics analysis identified a number of pollen-derived glycosides that were utilized in a species-specific manner. In particular, Lmel specialized in the utilization of flavonoids at the expense of simple sugars, which may explain why this species rapidly went extinct in presence of only simple sugars during the transfer experiments. While the importance of pollen-derived flavonoids in niche partitioning needs to be validated, the species-specific deglycosylation of these secondary plant compounds illustrates that the four species have different hydrolytic

capabilities that may also be involved in the cleavage of other carbohydrates. The metabolic specialization on plant glycans may be a common phenomenon in animal gut communities, as similar transcriptional changes have been described in other gut symbionts when the host diet was supplemented with specific plant glycans (*Sonnenburg et al., 2005*; *Zheng et al., 2019*).

In contrast to the species specific metabolism of glycoside, we observed few differences in the utilization of simple saccharides among the four species in our time-resolved GC-MS analysis. While this may seem surprising, theoretical work has established that resource preference for at least one substrate is sufficient to explain coexistence (*Meszéna et al., 2006*). Moreover, it is plausible that the four species utilize the same sugars, but extract them from different pollen-derived glycans, such as starch, hemicellulose, flavonoids, or other glycosylated secondary plant metabolites.

While this work focused on niche partitioning based on degradation of complex carbohydrates, it is noteworthy that all four *Lactobacillus* species engaged to a variable extent in co-fermentation of the carboxylic acids citrate and malate present in pollen. The two species, Lkul and Lhel, that upregulated citrate fermentation pathways in the presence of pollen also consumed citrate at the fastest rate. Citrate co-fermentation has been linked to competitive advantages in lactic acid bacteria, though whether the varying levels of co-fermentation contribute to colonization stability in this system remains an outstanding question (*Laëtitia et al., 2014*; *Magni et al., 1999*; *Jimeno et al., 1995*).

Previous work suggested that the large diversity of carbohydrate transport and metabolism functions in the accessory gene pool of *Lactobacillus* Firm5 is an adaptation to the pollen-based diet of the host and a consequence of the nutrient competition with closely related species (*Ellegaard and Engel, 2019*; *Ellegaard et al., 2019*). Our findings support this hypothesis and provide the first experimental evidence for a link between the coexistence of the four *Lactobacillus* species, the large diversity of carbohydrate metabolism functions in their genomes, and the pollen diet of the host. Moreover, these results suggest that dietary differences between host species or natural variation in pollen diversity influence the diversity of *Lactobacillus* Firm5 and could, for example explain why the Asian honey bee, *Apis cerana,* harbors only one species of this phylotype in its gut (*Ellegaard et al., 2020*).

However, we have only tested a single strain of each of the four species. Therefore, given the extensive genomic diversity within these species (*Ellegaard and Engel, 2019*), more work is needed to determine if the identified patterns of coexistence reflect stable ecological niches occupied by the four species or are rather the result of the specific strains selected for our experiments. In a recent study on pitcher plant microbiomes, it was shown that even strains that differ by only a few base pairs can have different ecological trajectories in communities and coexist over extended period of time (*Bittleston et al., 2019*). Expanding our approach to strains within species presents an exciting next step to understand at which level discrete ecological niches are defined in the bee gut and how diversity can be maintained in such ecosystems.

## Materials and methods

**Key resources table**

| Reagent type (species) or resource | Designation | Source or reference | Identifiers | Additional information |
|---|---|---|---|---|
| Strain, strain background (*Lactobacillus apis*) | Lapi | https://doi.org/10.1371/journal.pbio.2003467 | Genome ID: 2684622912 | |
| Strain, strain background (*Lactobacillus helsingborgensis*) | Lhel | https://doi.org/10.1371/journal.pbio.2003467 | Genome ID: 2684622914 | |
| Strain, strain background (*Lactobacillus melliventris*) | Lmel | https://doi.org/10.1371/journal.pbio.2003467 | Genome ID: 2684622913 | |
| Strain, strain background (*Lactobacillus kullabergensis*) | Lkul | https://doi.org/10.1371/journal.pbio.2003467 | Genome ID: 2684622911 | |

*Continued on next page*

*Continued*

| Reagent type (species) or resource | Designation | Source or reference | Identifiers | Additional information |
|---|---|---|---|---|
| Commercial assay or kit | QIAquick Gel Extraction Kit | Qiagen | #Cat 28706X4 | |
| Commercial assay or kit | Nucleospin RNA clean-up kit | Macherey-Nagel | #Cat 740903 | |
| Commercial assay or kit | Zymo-Seq RiboFree Total RNA Library kit | Zymo Research | #Cat R3000 | |
| Software, algorithm | R Studio software | R Studio (https://www.rstudio.com) | RRID:SCR_000432 | |
| Software, algorithm | Integrative Genomics Viewer | Integrative Genomics Viewer (https://software.broadinstitute.org/software/igv/) | RRID:SCR_011793 | |

## Culturing of bacterial strains

We used the following four bacterial strains of Lhel, Lmel, Lapi, and Lkul for our experiments: ESL0183, ESL0184, ESL0185, and ESL0186 (*Kešnerová et al., 2017*). All strains were precultured on solid De Man – Rogosa – Sharpe agar (MRSA) (supplemented with 2% w/v fructose and 0.2% w/v L-cysteine-HCl) from glycerol stocks stored at −80°C. MRSA plates were incubated for three days in anaerobic conditions at 34°C to obtain single colonies. Single colonies were inoculated into a liquid carbohydrate-free MRS medium (cfMRS; *O' Donnell et al., 2011*) supplemented with 4% glucose (w/v), 4% fructose (w/v), and 1% L-cysteine-HCl (w/v) and incubated at 34°C in anaerobic conditions without shaking.

## In vivo transfer experiments

Bacterial colonization stocks were prepared from overnight cultures by washing the bacteria in 1xPBS, diluting them to an $OD_{600} = 1$, and storing them in 25% glycerol at −80°C until further use. For colonization stocks containing all four species, cultures adjusted to an $OD_{600} = $ one were mixed at equal proportions. Microbiota-depleted bees were obtained from colonies of *Apis mellifera carnica* located at the University of Lausanne following the procedure described in *Kešnerová et al., 2017*. Colonization stocks were diluted ten times in a 1:1 mixture of 1xPBS and sugar water (50% sucrose solution, w/v) and 5 µL were fed to each bee using a pipette. Five days post-colonization, 10 rectums were dissected and homogenized in 1xPBS. An aliquot of each homogenized gut was used for CFU plating to enumerate the total bacterial load and for amplicon sequencing to obtain the relative abundance of each community member (see below). To serial passage the community through microbiota-depleted bees, the ten homogenized gut samples from the same treatment were pooled together and stored in 25% glycerol at −80°C until a new batch of microbiota-depleted bees was available. At the day of colonization, a frozen aliquot of the pooled gut homogenate was thawed, diluted ten times in a 1:1 mixture of 1xPBS and sugar water (50% sucrose solution, w/v), and fed to newly emerged microbiota-depleted bee as described above. This was repeated for a total of six serial passages. Throughout the experiments all bees were kept on either a sugar water or a sugar water/pollen diet according to the two dietary treatment. Food was provided ad libitum.

## In vitro transfer experiment

Each of the four strains was cultured in liquid medium overnight for about 16 hr as described above. The cultures were re-inoculated at an $OD_{600} = 0.3$ in fresh medium and let grow for another 4 hr at 34°C with shaking (700 rpm). Bacterial cells were then washed with 1xPBS, mixed in equal proportions, and inoculated at an $OD_{600} = 0.05$ in triplicates in 96-deep well plates (SIGMA) containing cfMRS medium supplemented with either 2% glucose (w/v), 10% pollen extract (v/v), or 10% pollen grains (v/v) in a final volume of 500 µL per well. Detailed information about pollen extract preparation can be found in the Supporting methods section of *Kešnerová et al., 2017*. Pollen grain solutions were prepared by adding 1.250 ml of $ddH_2O$ to 80 mg of pollen grains crushed with the bottom of a 15 mL falcon tube. The plates were incubated for 24 hr at 34°C under anaerobic conditions without shaking (300 rpm). After 24 hr of incubation, an aliquot of each sample was subjected

to CFU plating to enumerate the total bacterial load. Then, 1% of each culture (i.e. 5 µL) was transferred to a plate with fresh medium supplemented with the appropriate carbon sources and incubated again. These transfers were repeated 10, respectively, 20 times for the two independent experiments. After each transfer, cultures were washed once with 1xPBS and stored at −20°C for amplicon sequencing analysis. CFUs were counted after 24 hr and at the final transfer.

## Amplicon sequencing

The relative abundance of the four strains across all transfer experiments was obtained using amplicon sequencing of a 199 bp long fragment of a housekeeping gene encoding a DNA formamidopyrimidine-glycosylase which allows to discriminate the four strains from each other (*Ellegaard et al., 2019*).

For the in vitro transfer experiments, the PCR fragment was amplified from crude cell lysates. They were generated by mixing 5 µL of culture with 50 µL of lysis solution, containing 45 µL of lysis buffer (10 mM Tris- HCl, 1 mM EDTA, 0.1% Triton, pH 8), 2.5 µL of lysozyme (20 mg/ml, Fluka), and 2.5 µL of Proteinase K solution (10 mg/ml, Roth). The samples were incubated for 10 min at 37°C, for 20 min at 55 °C, and for 10 min at 95 °C, followed by a short spin before preparing the PCR (1 min, 1500 rpm). For the in vivo transfer experiment, DNA was isolated from the homogenized gut samples using the hot phenol protocol used in *Kešnerová et al., 2017*.

To amplify the gene fragment and to add the Illumina barcodes and adapters, the two-step PCR strategy published in *Ellegaard et al., 2019* was used. For the first PCR, 5 µL of DNA or 5 µL of cell lysate were mixed with 12.5 µL of GoTaq Colorless Master Mix (Promega), 1 µL of forward and reverse primer (5 µM, see *Supplementary file 1*) and 5.5 µL of Nuclease-free Water (Promega). The PCR I was performed as follows: initial denaturation (95°C – 3 min), 30 times denaturation-annealing-extension (95°C – 30 s, 64°C – 30 s, 72°C – 30 s), final extension (72 °C – 5 min). To purify the amplicons, 15 µL of PCR product were mixed with 5 µL of a 5X Exo-SAP solution (15% Shrimp Alkaline Phosphatase – 1000 U/ ml – NEB, 10% Exonuclease I – 20,000 U/ ml – NEB, 45% glycerol 80% and 30% dH2O). and incubated for 30 min at 37°C and for 15 min at 80°C. For the second PCR reaction, 5 µL of purified PCR products were mixed with the same reagents as before. The PCR program was the same as above with the exception that the annealing temperature was set to 60°C and the denaturation-annealing-extension steps were repeated for only eight times. The barcoded primers are listed in *Supplementary file 1*. The amplicons of the second PCR were purified using the Exo-SAP program as described above.

To prepare the sequencing of the amplicons, DNA concentrations were measured using Quant-iT PicoGreen for dsDNA (Invitrogen). Each sample was adjusted to a DNA concentration of 0.5 ng/µL and 5 µL of each sample were pooled together. The pooled sample was loaded on a 0.9% agarose gel and gel-purified using the QIAquick Gel Extraction Kit (Qiagen) following the manufacturer's instructions. The purified DNA was prepared for sequencing following the Illumina MiniSeq System Guide for 'denaturate and dilute libraries' and then loaded on a Illumina MiniSeq Mid Output Reagent Cartridge using the correspondent MiniSeq flow cell. Illumina reads were demultiplexed by retrieving the unique barcodes of the different samples and quality-filtered using Trimmomatic (Trimmomatic-0.35) (LEADING:28 TRAILING: 29 SLIDING WINDOW:4:15 MINLEN:90). Each forward and reverse read pair was assembled using PEAR (-m 290 n 284 j 4 -q 26 v 10 -b 33) (*Zhang et al., 2014*), and the assembled reads were assigned to the different strains based on base pair positions with discriminatory SNP. See details in Supplementary material.

To obtain absolute abundance data for each strain, we combined the relative abundance data from the amplicon sequencing with CFU counts obtained from plating homogenized bee guts in the case of the in vivo experiments (see above) or by carrying out qPCR with *Lactobacillus*-specific primers as described in *Kešnerová et al., 2017* in the case of the in vitro co-culture experiments (*Supplementary file 1*, *Supplementary file 4*). For the in vitro transfer, the stability of the four-species community over time was calculated using the codyn R package (*Hallett et al., 2020*).

## RNA extraction and sequencing

For the in vivo RNA sequencing, microbiota-depleted bees were colonized with the four species community as described above and fed with either sugar water and pollen grains or with sugar water only. After 5 days of colonization, the rectums of five bees per treatment (all kept in the same cage)

were dissected and snap-frozen in liquid nitrogen in separate tubes containing glass beads (0.1 mm dia. Zirconia/Silica beads; Carl Roth). For RNA extraction, the tissue samples were suspended in 1 mL of TE buffer and homogenized using a bead beater (45 m/s, 6 s). Then, 200 µL of an ice-cold STOP solution (95% v/v ethanol, 5% v/v Aqua phenol [Roth]) was added to 1 ml of homogenate and snap-frozen again in liquid nitrogen. Tubes were then thawed on ice and a previously developed hot phenol RNA extraction protocol was followed (*Sharma et al., 2010*). For the in vitro RNA sequencing, bacterial strains were cultured in triplicates in cfMRS supplemented with either 1% w/v glucose or 1% w/v pollen extract. After 16 hr of growth, 200 µL of STOP solution was added to 1 mL of culture followed by the same steps as described above.

After the precipitation step, samples were treated with DNaseI (NEB) to degrade DNA. RNA samples were purified using Nucleospin RNA clean-up kit (Macherey-Nagel) following the manufacturer's instructions. RNA was eluted in RNase free-water and stored at −80°C until further use. RNA concentration and quality were assessed using Nanodrop (ThermoFisher Scientific), Qubit (ThermoFisher Scientific, RNA – High Sensitivity reagents and settings) and Bioanalyzer (Agilent). High-quality RNA samples were selected to prepare RNA libraries. For the in vivo RNA sequencing, libraries were prepared using the Zymo-Seq RiboFree Total RNA Library kit (Zymo Research). The libraries were sequenced by the GTF facility of the University of Lausanne using HiSeq 4000 SR150 sequencing (150 bp reads) (Illumina). For the in vitro RNA sequencing, libraries were prepared following the protocol developed by *Avraham et al., 2016*. Libraries were then prepared for sequencing following the Illumina MiniSeq System guide for denaturate and dilute libraries. Libraries were sequenced using the Illumina MiniSeq technology using High Output Reagent Cartridges (150 bp reads) and MiniSeq flow cells.

## RNA sequencing analysis

For the in vitro samples, raw reads were demultiplexed using a script provided by Dr. Jelle Slager (Personal communication) For the in vivo samples, the reads were already demultiplexed by the sequencing facility. For both experiments, the reads were trimmed with Trimmomatic (Trimmomatic-0.35) (LEADING:30 TRAILING: 3 SLIDING WINDOW:4:15 MINLEN:20). The quality of the reads was checked using FASTQC (http://www.bioinformatics.babraham.ac.uk/projects/fastqc/). For the in vivo samples, trimmed reads were sorted with sortmerna-4.2.0 to select only the non-rRNA reads for the downstream analysis. Reads were mapped onto the genomes of the selected strains (*Ellegaard and Engel, 2018*) (Lapi, Lhel, Lmel, and Lkul) using Bowtie (bowtie2-2.3.2). Gene annotations for the four genomes were retrieved from IMG/mer (*Chen et al., 2021*). Mapped reads were quality filtered for the alignment length (CIGAR > 100 bp) and for the allowed mismatches in the sequence (NM = 0–1). Quality filtered reads were then quantified using HTseq (Version 0.7.2). Differential gene expression between samples cultured in pollen extract and samples cultured in glucose, and between mono-cultures and co-cultures, was calculated using the R package EdgeR (*Robinson et al., 2010*). Counts per million were calculated and only genes with at least one count per million were used for the analysis. EdgeR fits negative binomial models to the data. The counts were normalized for RNA composition by adjusting the $log_2FC$ according to the library size, and the quantile-adjusted conditional maximum likelihood (qCML) method was used to estimate the common dispersion and the tag-wise dispersion. Finally, the differential gene expression was determined using the exact test with a false discovery rate (FDR) <5%. COG annotations were obtained from IMG/mer, and the enrichment analysis for COG categories tested using the Fisher's exact test. Transcripts per million (TPM) were visualized using the Integrated Genome Browser software (*Freese et al., 2016*).

## Untargeted metabolomics

Metabolites were extracted from liquid cultures supplemented with 10% (w/v) pollen extract at the inoculation time and after 16 hr of incubation at 34°C. For each liquid culture sample, 300 µL was collected and centrifuged (20,000 g, 4°C, 30 min), then 200 µL supernatant was transferred to a new tube and stored at −80°C. After collection of all samples, they were prepared for metabolomics analysis. The samples were thawed on ice and centrifuged again (20,000 g, 4°C, 5 min), then diluted 10 times with ddH₂O. For metabolomics analysis, 25 µL of each diluted sample was sent in a 96-well plate on dry ice to the laboratory of Prof. Uwe Sauer for analysis (ETH Zürich, Switzerland). Three replicates of a pollen-extract dilution series (10 serial 2x dilutions) as well as undiluted pollen-extracts

and water used for performing the dilution series were included in the metabolomics analysis. Because of the insolubility of flavonoid aglycones in a water matrix, metabolites from liquid cultures supplemented with rutin were extracted using a methanol-extraction protocol at the time of inoculation and after 16 hr of growth by adding 200 µL of methanol pre-cooled to −20°C to 100 µL of culture. Tubes were vortexed thoroughly and incubated for 5 min (4°C, shaking 14,000 rpm). Samples where then incubated at −20 °C for 1 hr and centrifuged (20'000 g, 5 min). A total of 200 µL of the supernatant was transferred to a new tube and diluted 10 times in 70% methanol and 25 µL of each diluted sample was sent to Zürich in Eppendorf tubes sealed with parafilm on dry ice. For untargeted metabolomics analysis, each sample was injected twice (technical replicate) into an Agilent 6550 time-of-flight mass spectrometer (ESI-iFunnel Q-TOF, Agilent Technologies) as detailed in *Kešnerová et al., 2017*. In brief, m/z features (ions) were annotated by matching their accurate mass-to-sum formulas of compounds in the KEGG database accounting for deprotonation (-H$^+$). Alternative annotation can be found in *Supplementary file 10*. When available, metabolites categories were assigned to ions based on KEGG ontology.

Metabolomics data analysis was carried out using R version 3.6.3. Variation of raw ion intensities obtained from untargeted metabolomics analysis for the two technical replicates was determined by assessing the correlation between ion intensities of the respective technical replicates. Then, mean ion intensities of technical replicates were calculated. Time point comparisons (T = 0 hr vs T = 16 hr) were performed using t-tests with Benjamini-Hochberg (BH) correction for multiple testing. log$_2$FC values between the two time-points were calculated with respect to the mean intensity in the T0 time point. To identify pollen-derived ions, and distinguish them from background originating from culture medium and experimental noise, the ion intensities of the pollen dilution series were plotted for each ion and the R (2) of the obtained linear fit was extracted. In addition, we calculated the log$_2$FC difference between undiluted pollen and water. The R (2) values were then plotted against the log$_2$FC values, and stringent thresholds (R$_2$ > 0.75 and log$_2$FC > 2) were chosen to discriminate ions that are likely pollen-derived (*Figure 6—figure supplement 3*). All ions were included for downstream analysis (e.g. PCA) and then they were discriminated between pollen-derived and non-pollen-derived.

## Semi-targeted metabolomics via GC-MS

Soluble metabolites were extracted from liquid cultures supplemented with 10% pollen extract (w/v) at the inoculation time and after 8, 16, and 24 hr of incubation. For each liquid culture sample, 300 µL was collected and centrifuged (15,000 g, 4°C, 15 min). Then, 200 µL was transferred to a new tube, snap-frozen in liquid nitrogen, and stored at −80°C. Once that all the samples were collected, soluble metabolites were extracted. To extract soluble metabolites, tubes were thawed on ice, and 75 µL of sample was combined with 5 µL of 20 mM internal standard (norleucine and norvaline, (Sigma-Aldrich) and U-$^{13}$C$_6$ glucose [Cambridge Isotope laboratories]). A volume of 825 µL of cold methanol:water:chloroform (5:2:2) solution was added to the sample and vortexed for 30 s. The tubes were incubated at −20°C for 90 min and vortexed 2x for 30 s during the incubation. Tubes were centrifuged for 5 min at 10,000 g at 4°C. The supernatant was removed and extraction was repeated using 400 µL of ice cold chloroform:methanol (1:1), tubes were vortexed and left on ice for 30 min. Tubes were centrifuged 5 min at 8000 rpm at 4°C and the liquid phase was transferred to the previous extracted aqueous phase. A total of 200 µL of water was added and tubes were centrifuged 5 min at 8000 rpm. The aqueous phase was transferred to a 2 mL microcentrifuge tube. The aqueous extract was dried using a vacuum concentrator at ambient temperature overnight (Univapo 150 ECH vacuum concentrator centrifuge). Once dried, the samples were dissolved in 50 µL of 20 mg/ml methoxyamine hydrochloride in pyridine for 1.5 hr at 33°C followed by derivatization with N-Methyl-N-(trymethylsolyl)trifluoroacetamide (MSTFA, Sigma Aldrich) for 2 hr at 35°C.

Aliquots (1 µL) were injected on an Agilent 8890/5977B GC-MSD. The samples were injected in split mode (20:1) with an inlet temperature of 250°C. The VF-5ms (30 m x 250 µm x 0.25 µm) column was held initially at 125°C for 2 min, ramped at 5°C / min to 250°C, ramped at 15°C to 300°C, and held for 5 min. The MS was run in full scan mode (50–500 m/z) at a speed of 5 Hz. Peaks from the total ion chromatogram (TIC) were identified by matching retention times and spectra to an in-house library that was built by comparing selected T=0h and T=24h samples against the NIST library, as well as our library of analytical standards. Compounds are noted as either confirmed with our own standards, or the best match and associated matching factor against the NIST library are reported

(*Supplementary file 10*). Peaks were picked and integrated using the Agilent MassHunter Quantitative Analysis software. Peak areas were normalized to the internal standards. The data were processed using R version 3.6.3 and mean intensities and $\log_2$FC between time-points were calculated as described above for the untargeted metabolomics analysis.

### Analysis-code and data availability

The complete custom code for all the analyses is available on GitHub: (https://github.com/silviabrochet/Brochet_2021_eLife, copy archived at swh:1:rev:237a27f757296372f0333d298dfb7c765686fe03; *Brochet, 2021*). The amplicon sequencing data and the RNA sequencing data are available under the NCBI Bioproject PRJNA700984 and the GEO record GSE166724. All differential expression analysis results of this study are included in *Supplementary file 10*.

## Acknowledgements

We thank Kirsten Ellegaard for the critical reading of the manuscript and discussions on the experimental design. We also thank Sara Mitri for providing feedback on the manuscript. We are grateful to Simon Maréchal for his help with the transfer experiments. This project was funded by the ERC-StG 'MicroBeeOme' (grant 714804), the Swiss National Science Foundation (grant 31003A_160345 and 31003A_179487), the Human Frontier Science Program Young Investigator (grant RGY0077/2016) and the NCCR Microbiomes.

## Additional information

### Funding

| Funder | Grant reference number | Author |
|---|---|---|
| H2020 European Research Council | 714804 | Silvia Brochet<br>Philipp Engel |
| Swiss National Science Foundation | 31003A_179487 | Andrew Quinn<br>Nicolas Neuschwander<br>Philipp Engel |
| Swiss National Science Foundation | 51NF40_180575 | Philipp Engel |
| Swiss National Science Foundation | 31003A_160345 | Philipp Engel |
| Human Frontier Science Program | RGY0077/2016 | Philipp Engel |

The funders had no role in study design, data collection and interpretation, or the decision to submit the work for publication.

### Author contributions

Silvia Brochet, Conceptualization, Visualization, Methodology, Writing - original draft, Writing - review and editing; Andrew Quinn, Conceptualization, Data curation, Methodology, Writing - review and editing; Ruben AT Mars, Resources, Investigation, Methodology, Writing - review and editing; Nicolas Neuschwander, Investigation; Uwe Sauer, Resources, Writing - review and editing; Philipp Engel, Conceptualization, Software, Supervision, Funding acquisition, Visualization, Writing - original draft, Project administration, Writing - review and editing

### Author ORCIDs

Silvia Brochet (iD) https://orcid.org/0000-0002-6443-185X
Andrew Quinn (iD) https://orcid.org/0000-0003-1401-1053
Uwe Sauer (iD) https://orcid.org/0000-0002-5923-0770
Philipp Engel (iD) https://orcid.org/0000-0002-4678-6200

Decision letter and Author response
Decision letter https://doi.org/10.7554/eLife.68583.sa1
Author response https://doi.org/10.7554/eLife.68583.sa2

## Additional files

### Supplementary files

• Supplementary file 1. List of primers used in this study: the sample-specific barcodes used in the primers for the second PCR of the amplicon sequencing are highlighted in gray.

• Supplementary file 2. COG functional categories.

• Supplementary file 3. Bacterial abundance data (CFUs).

• Supplementary file 4. Amplicon sequencing processed data.

• Supplementary file 5. RNA sequencing processed data, statistical analysis results (enrichment tests), transcript per million data.

• Supplementary file 6. RNA sequencing processed data, statistical analysis results (enrichment tests), transcript per million data.

• Supplementary file 7. RNA sequencing processed data, statistical analysis results (enrichment tests), transcript per million data.

• Supplementary file 8. RNA sequencing processed data, statistical analysis results (enrichment tests), transcript per million data.

• Supplementary file 9. RNA sequencing processed data, statistical analysis results (enrichment tests), transcript per million data.

• Supplementary file 10. Metabolomics analysis data.

• Supplementary file 11. Differential expression analysis results.

• Transparent reporting form

### Data availability

The amplicon sequencing data and the RNA sequencing data are available under the NCBI Bioproject PRJNA700984 and the GEO record GSE166724 respectively. All data generated or analysed during this study are included in the manuscript and supporting files. Bacterial abundance data (CFUs) are included into Supplementary file 3, amplicon sequencing processed data are included into Supplementary file 4, RNA sequencing processed data, statistical analysis results (enrichment tests) and transcript per million data are included into Supplementary file 5–9, metabolomics analysis data are included into Supplementary file 10. All differential expression analysis results of this study are included in Supplementary file 11.

The following datasets were generated:

| Author(s) | Year | Dataset title | Dataset URL | Database and Identifier |
|---|---|---|---|---|
| Brochet S, Quinn Q, Mars RAT, Neuschwander N, Sauer U, Engel P | 2021 | Lactobacillus Firm5 amplicon sequencing | https://www.ncbi.nlm.nih.gov/bioproject/PRJNA700984 | NCBI BioProject, PRJNA700984 |
| Brochet S, Quinn Q, Mars RAT, Neuschwander N, Sauer U, Engel P | 2021 | Lactobacillus Firm5 in vivo and in vitro RNA sequencing | https://www.ncbi.nlm.nih.gov/geo/query/acc.cgi?acc=GSE166724 | NCBI Gene Expression Omnibus, GSE166724 |

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
