## [Decision Letter]

**Acceptance summary:**

Brochet et al., investigate how closely related microbial symbionts coexist in the gut as stable communities. To address this question, the authors use an elegant model, relying on honeybees colonized with a defined bacterial community. They provide compelling empirical evidence that a nutritionally complex diet, together with microbial metabolic diversity, play a key role, enabling species partitioned resource utilization and co-existence of closely related honeybee gut microbiota.

**Decision letter after peer review:**

Thank you for submitting your article "Niche partitioning facilitates coexistence of closely related gut bacteria" for consideration by *eLife*. Your article has been reviewed by three peer reviewers, including Karina Xavier as the Reviewing Editor and Reviewer #1, and the evaluation has been overseen by Christian Rutz as Senior Editor.

The reviewers have discussed their reviews with one another, and the Reviewing Editor has drafted this decision letter to help you prepare a revised submission.

Essential Revisions (for the authors):

In general, all three reviewers found the work exciting and solid, and thus suitable for publication in *eLife*. Below is a list of essential revisions to help improve the clarity and presentation of the work. Additionally, you can find at the end of this letter, for context, the separate "recommendations for the authors" written by the individual reviewers. We hope you will use this information to improve your manuscript, but you do not need to reply to the comments raised by the reviewers -- you will only need to reply to the following essential points summarized here:

1) The experimental design chosen for the in vitro cultures is not clear. It is not clear why a sucrose solution (SW) was used for the in vivo studies and glucose was used for the in vitro studies and why a "simple sugar + pollen" group was not included in the in vitro study. Since we believe it is unlikely to affect the major findings of the paper where comparisons of simple sugars and more complex diets are made, we are not asking for additional experiments. But we think it is important that you explain your rationale, and discuss if and how these choices may impact your results.

2) You do not explore much the mechanisms that can explain the lack of persistence observed under the simple diet. We acknowledge that a comprehensive understanding of the mechanisms involved in the simple diet are beyond the scope of the present paper, given that here you provide a clear proposal of the more interesting problem, that is the co-existence and persistence in the complex diet. However, we think that the paper could improve with an added analysis and/or discussion with the current data of possible mechanisms involved (see major comment 1 of Reviewer #1, point 4 of Reviewer #2, and point 2 of Reviewer #3).

3) Please justify more clearly your choice of study organisms, and in the discussion, comment on how your findings can be extended to the related bumble bee gut microbiome (see point 5 of Reviewer #2, and point 1 of Reviewer #3).

4) It is important to make available the genome-wide data for the RNA-seq results, not just for selected genes.

*Reviewer #1 (Recommendations for the authors):*

1. The current study does not explore what happens in the simplified diet both in vitro and in the Bee gut. The authors propose in the discussion that in the presence of few and simple carbon sources (sugars) there is competition for nutrients and competitive exclusion is driving extinction of most species and domination of a single species. However, there is not much experimental support for this conclusion. Can this be explained by competitive exclusion? Is there transcriptomics (or genomic data) in this study that supports this possibility? If the data cannot provide strong hypothesis for competitive exclusion by nutrition competition in the simplified environment the authors should discuss other possible mechanisms that could explain such phenotypes in the simplified diets.

2. One point that it is not address is the number of calories provided by the different diets. Could it be that the simplified diet is also lower in calories (and thus more a limiting) and that differences in calories (as opposed to the different number of nutrient sources) could contribute to the results observed?

3. Regarding experimental design it is not clear why a sucrose solution (SW) was used for the in vivo studies and Glucose was used for the in vitro studies. Additionally, in vitro the results with pollen grains were better in promoting persistence than the Pollen extracts, but the follow up studies were performed with pollen extract that enables persistence but not as well. The authors should explain the rationale for these choices and discuss if these choices may or may not have impact on the results obtained.

*Reviewer #2 (Recommendations for the authors):*

Overarching comments:

1. My ability to make specific comments was hindered by the lack of line numbers and page numbers in the merged document. Perhaps these fell out of the compiled version, but if not, I urge the authors to include page numbers and line numbers in their future submissions.

2. I found it confusing that the bee diet conditions contrasted were pollen+sugar water or just sugar water, but the culture conditions contrasted were pollen (extract or granulated) or sugar water. I think that this may impact some statements that the authors are making regarding the comparison between in vitro and in vivo. I've noted specific cases in my comments below but wanted to reiterate this overarching concern here. Could the authors comment on the decision to exclude the conditions pollen only bee diet and/or in vitro pollen+sugar water from this study?

Introduction

3. You are covering a lot of ground in the introduction- one thing that got lost for me until the second read through was that I think you specifically choose the Firm5 subset of the bee gut microbiota because of close taxonomic relatedness. Consider motivating the paragraph beginning "One of the most prevalent phylotypes…" so that this is clearer.

4. I think that you have very well thought out reasons for choosing the species and strains for this study, and I think that it would strengthen the manuscript to spend a little more time on this in the introduction. I'm not very familiar with the bee gut system, and my first impression was to wonder why so few strains were tested, and why the strains chosen were representative/good choices. Some of this you mentioned in the discussion- I think it would strengthen the manuscript to present these details sooner. Specific questions that I think need to be addressed in the introduction are: Why were *Lactobacilli* chosen specifically, over other major phylotypes? What fraction of the total bee gut diversity does this phylotype represent? Is there generally only one strain of a species present in the bee gut? How were strains chosen as representatives?

Section: The coexistence of four related *Lactobacillus* species in the honey bee gut depends on the host diet

5. I think it would help the reader to unpack the idea in your summary paragraph. This is the first place where niche partitioning is mentioned (partitioning is referred to in the abstract), and you don't define the term. I think the link between niche partitioning and your results is that pollen is a complex resource, and sugar water is not. From consumer-resource models, one would expect that there would be only one dominant competitor on one resource. If multiple resources are available, then one would expect that the community would be as diverse as the # of resources. Since pollen has >4 resources, coexistence is possible if strains partition the resources they consume and do not compete directly.

6. I am *not* suggesting more experiments here, but have you considered seeing whether you could support the 4-member community on sugars found in pollen and detected as consumed by the metabolomics? Showing that this could work in a defined environment with these strains would be a nice (although not surprising) extension of the resource partitioning idea.

Section: in vitro co-culture experiments recapitulate the diet-dependent coexistence of the four *Lactobacillus* species

7. In culture, you observe higher yield for all four strains on glucose than on pollen or pollen extract, but you see the opposite in the bee gut. Have you followed up on why this might be? I ask because you are using yield as a proxy for 'niche space', and I think a fundamental assumption there is that resources that define the niches are limiting. If in culture a different resource is limiting (let's say resource B), then all of your strains might reach the same yield and stop growing due to resource B limitation before they have consumed all of the glucose or pollen. One way to determine whether glucose/ pollen limits yield would be to grow your strains on different concentrations of glucose/ pollen- if yield is dependent on the resource you are adding, you should see a linear relationship between the yield and the amount of glucose/pollen. I think that establishing that pollen or glucose is truly growth limiting in these experiments is critical if you want to make the statement that "portioning of pollen-derived nutrients is sufficient for enabling coexistence".

8. Your experiments in bees use either sugar water or sugar water+pollen grains. But the experiments in this section use either pollen grains, pollen extract, or sugar. It would be a clearer comparison with the in vivo experiments if you showed coexsistence on pollen+sugar water. Without comparing the same conditions, I don't think that you can conclude from this section that "In summary, these findings show that the dietary-dependent coexistence of the four *Lactobacillus* species observed in vivo can be recapitulated in vitro in a simple co- culture experiment, suggesting that the partitioning of pollen-derived nutrients is sufficient for enabling coexistence." I'd encourage the authors to re-phrase the summary and header for this section: I think that what is shown is that strains coexist on pollen but not on glucose, not that the diet-dependent coexistence has been recapitulated in vivo.

Section: The four *Lactobacillus* species upregulate divergent carbohydrate transport and metabolism functions during gut colonization in the presence of pollen

9. I find the down-regulation of amino-acid metabolism/transport genes in the SW+pollen condition to be interesting but counter-intuitive. Pollen is supplying dietary amino acids, whereas SW alone does not. Sometimes transcription becomes elevated when substrates are scarce, is there evidence in the literature that this could be the case?

10. Are the sugars transported/catabolized by the encoded COG G genes present in pollen?

11. I'm surprised that you didn't encounter problems with reads mapping to multiple reference genomes with HTSeq. The taxa share 85% gANI. I'd expect there to be some frequency of 100-mers sharing 1 SNP, which I understand from the results was the threshold for mapping reads to the genomes. I don't see any discussion of this in the text. Could you comment on what controls were done to explore the frequency of cross-mapping of reads?

12. Figure 4. My understanding of GH families is that they can be quite broad. For example, the GH16 family encompasses 15 different types of enzyme, not just the one cited in the figure. Could the authors provide more detailed information about why specific activities were attributed the GH families highlighted in the figure panels?

Section: Transcriptional responses to pollen are concordant in vivo and in vitro

13. Echoing my comments in the section about in vitro co-culturing, I'm wondering why the comparison was not glucose vs. glucose+pollen. In the host, when there is pollen available glucose is also available. Because of this, some of the differences between the in vitro pollen condition and the in vivo pollen+glucose condition might be transcripts that differentially expressed in response to glucose AND pollen, and not transcripts that are only expressed in the host. Because of this, I don't agree with the interpretation of the data that the transcriptional responses are concordant, and I think that the interpretation of differential expression to suggest that "genes are specifically expressed in vivo" is overly strong.

14. I think that these data can be used to make the overall point that the authors are presenting: the four strains do express different sugar-catabolism related genes when grown on pollen. Rather than comparing directly to the RNA-Seq from the host, I'd encourage the authors to focus instead on what their data show about the regulation of COG-G category genes in response to nutrient source. Specifically:

a. How does gene expression change between mono- and co-culture on the same carbon source?

b. If the strains differentially expressing genes when they are co-cultured, is a more diverse profile of COG-genes expressed, or do the strains tailor their expression?

c. I think there is only one direct comparison: co-culture communities on glucose. Are the COG genes differentially expressed the co-culture glucose community in vitro also the ones that are differentially expressed in the context of the in vivo community grown on glucose?

15. Supplementary Figure 3 indicates that some host samples profiled by RNASeq did not cluster by mds because of low read mapping. Please clarify if these samples were included in other analyses involving RNASeq data or if they were removed. If they were included, please comment on the rational for inclusion given low reads and lack of clustering.

Section: Metabolomics analysis reveals differences in flavonoid and sugar metabolism across the four *Lactobacillus* species

16. In studies of non-host associated microbial communities, an observation is that there is strain-level heterogeneity with respect to the profile of carbon sources consumed. From the metabolomics data, it looks as though some strains don't consume some carbon sources. Do you know whether all 4 strains can grow on the different carbon sources that you find by metabolomics in pollen, in particular the carbon sources that are most abundant in pollen?

Discussion:

17. I don't think that the comparison to hydra is fair- hydra microbiota assemble on the host epithelium, not a gut lumen.

18. You note that the primary site for colonization of *Lactobacillus* in the bee gut (the rectum) is known, and that other microbial taxa of the bee gut microbiota display different tissue tropism. I think that this is a really interesting point. Would we expect the dynamics of bee gut taxa with a closer host association to be different? Was the choice of strains motivated by the tissue tropism?

19. I'm surprised that you don't see evidence of cross-feeding on the complex resource. Do you think that this is due to the taxonomic similarity of the isolates (do you think crossfeeding is restricted to other phylotypes), or the small # used in this study, or that the community doesn't become limited for the resources in pollen? Can you differentiate among these possibilities with your data?

---

## [Author Response]

1) The experimental design chosen for the in vitro cultures is not clear. It is not clear why a sucrose solution (SW) was used for the in vivo studies and glucose was used for the in vitro studies and why a "simple sugar + pollen" group was not included in the in vitro study. Since we believe it is unlikely to affect the major findings of the paper where comparisons of simple sugars and more complex diets are made, we are not asking for additional experiments. But we think it is important that you explain your rationale, and discuss if and how these choices may impact your results.

It is true that the in vitro and in vivo nutrient conditions were not identical, and retrospectively it would have been more consistent to compare pollen versus glucose both in vitro and in vivo. However, we need to keep in mind that whatever we would have fed to the bees, would have been pre-digested and partially absorbed by the host tissue in the midgut before it reaches the bacteria in the hindgut. For example, sucrose is cleaved by the bee into glucose and fructose and absorbed in the midgut. Consequently, the in vivo and in vitro conditions will never be the same even if we would have used identical treatments. Despite these inconsistencies between the in vivo and in vitro treatments, we agree with the reviewers that the experimental design is still valid to test the main hypothesis of our study, i.e. a complex, pollen-based diet facilitates coexistence, while the provisioning of simple sugars leads to the dominance of a single strain independent of the environment (culture tube versus host).

We have revised the discussion to acknowledge that the different sugar conditions may explain the differences in the relative abundance of community members between the in vitro and in vivo experiments. This can be found in the revised manuscript on lines 402-422.

2) You do not explore much the mechanisms that can explain the lack of persistence observed under the simple diet. We acknowledge that a comprehensive understanding of the mechanisms involved in the simple diet are beyond the scope of the present paper, given that here you provide a clear proposal of the more interesting problem, that is the co-existence and persistence in the complex diet. However, we think that the paper could improve with an added analysis and/or discussion with the current data of possible mechanisms involved (see major comment 1 of Reviewer #1, point 4 of Reviewer #2, and point 2 of Reviewer #3).

We have addressed this point by adding a section in the discussion where we explain in more detail why we hypothesize that the lack of persistence of the four strains in the presence of the simple nutrients is due to competitive exclusion. The key point here is that each strain individually colonizes the bee gut, respectively grows in the culture tubes, in presence of the simple nutrients. However, when co-cultured only one strain persists in the simple nutrient condition, while in the presence of pollen all strains persist. The difference between these two conditions is the presence of pollen, i.e. a nutrient-rich substrate. The competitive exclusion principle states that two individuals can’t coexist if they compete for the same resource. The strain with the slightest growth advantage over the others will win, be it because it can utilize the available nutrient faster or actively inhibit the others in their growth. This is what our data suggests. We cannot see any other possible mechanism explaining this data.

The section in the discussion is found on line 423-435.

3) Please justify more clearly your choice of study organisms, and in the discussion, comment on how your findings can be extended to the related bumble bee gut microbiome (see point 5 of Reviewer #2, and point 1 of Reviewer #3).

We have addressed this point by restructuring the part of the introduction dedicated to *Lactobacillus* Firm5 on line 75-89. In brief, we chose this phylotype because (i) it is one of the most abundant phylotype of the honey bee gut microbiota, (ii) it is a very diverse phylotype that diverged into four different species which are different in their carbohydrate-utilization related gene content, (iii) the four species were shown to consistently coexist in vivo in conventional bees.

As for the bumble bee, or other social bee microbiota, we don’t know yet how many different lineages of *Lactobacillus* Firm5 coexist in other bees. Two different lineages of Firm5 have been found in bumble bees, but it is unclear if they coexist, neither is it known if these are the only two lineages associated with bumble bees. We feel that there is currently too little data to discuss how our results would relate to diversity found in other bee species. However, we added a sentence in the discussion which states that given the fact that the coexistence is based on pollen nutrients, it follows that dietary differences are likely to influence the diversity of *Lactobacillus* Firm5 in divergent host species. We give as an example *Apis cerana*, which harbors only one species of *Lactobacillus* Firm5. See line 479-488.

4) It is important to make available the genome-wide data for the RNA-seq results, not just for selected genes.

We have addressed this point by adding all differentially regulated genes in the supplementary dataset 9, in addition to the subset of significant ones. Please, note that in the GEO submission we included all the raw data used for the analysis and the scripts can be found in the GitHub link indicated in the text.